# Molecular “Yin-Yang” Machinery of Synthesis of the Second and Third Fullerene C_60_ Derivatives

**DOI:** 10.3390/mi16070770

**Published:** 2025-06-30

**Authors:** Djuro Lj. Koruga, Lidija R. Matija, Ivana M. Stanković, Vladimir B. Pavlović, Aleksandra P. Dinić

**Affiliations:** 1NanoLab, Biomedical Engineering, Faculty of Mechanical Engineering, University of Belgrade, 11120 Belgrade, Serbia; lmatija@mas.bg.ac.rs (L.R.M.); imileusnic@mas.bg.ac.rs (I.M.S.); adinic@mas.bg.ac.rs (A.P.D.); 2TEM Laboratory, Faculty of Agriculture, University of Belgrade, 11000 Belgrade, Serbia; vlaver@agrif.bg.ac.rs

**Keywords:** hydrogen bonds, water, DNA, proteins, fullerene C_60_ derivatives, FTIR, TEM, AFM/MFM, nano-physical “Yin-Yang” machinery

## Abstract

To overcome the negative effects of the biochemical application of nano-substances in medicine (toxicity problem), using the example of fullerene C_60_’s first derivative (fullerenol, FD-C_60_), we show that their biophysical effect is possible through non-covalent hydrogen bonds when around FD-C_60_ water layers are formed. SD-C_60_ (Zeta potential is −43.29 mV) is much more stable than fullerol (Zeta potential is −25.85 mV), so agglomeration/fragmentation of the fullerol structure, due to instability, can cause toxic effects. When fullerol in solution was exposed to an oscillatory magnetic field with *Re* (real) part [250/−92 mT, H(*ωt*) = *Acos*(ωt)], water layers around FD-C_60_ (fullerenol) are formed according to the Penrose process of 3D tiling formation, and the second derivative, SD-C_60_ (or 3HFWC), is self-organized. However, when *Im* (imaginary) part [250/−92 mT, H(*ωt*) = *Bisin* (ωt)] of the external magnetic field is applied in addition to SD-C_60_, ordered water chains and bubbling of water (“micelle”) are formed as a third derivative (TD-C_60_). Fullerol (FD-C_60_) interacts with biological structures biochemically, while the second (SD-C_60_) and third (TD-C_60_) derivatives act biophysically via non-covalent hydrogen bond oscillation. SD-C_60_ and TD-C_60_ significantly increased water solubility and reduced toxicity. The paper explains the synthesis of SD-C60 and TD-C60 from FD-C60 (fullerol) as a precursor by the influence of an oscillatory magnetic field (“Yin-Yang” principle) on hydrogen bonds in order to create water layers around fullerol. Examples of biomedical applications (cancer and Alzheimer’s) of this synergetic complex are given. This study shows that the “Yin-Yang” machinery, based on the nanophysics of C_60_ molecules and non-covalent hydrogen bonds, is possible. The first attempt has been composed to synthesize nanomaterial for biophysical vibrational nanomedicine.

## 1. Introduction

The first potential of nano-machinery in medical and pharmaceutical uses is through its application in the delivery of existing therapeutic and diagnostic agents. Today’s nanopharmacology deals with the application of nanoscience and nanotechnology based on new pharmacological principles with the aim of increasing therapeutic effectiveness and reducing side effects, as well as achieving targeted delivery of the drug to specific locations in a controlled manner. Nanocarriers, such as liposomes, nanoparticles, nanostructured lipid carriers, polymer micelles, and polymer conjugates of drugs with a size of 200 nm show promising effectiveness in the field of medicine. However, there is a lack of data on clinical studies and the usefulness of DDS (drug delivery system) in patients. Pharmacologists must be involved in the investigation of pharmacokinetics and pharmacodynamics of DDS if the products have reached clinical use [1,2,3].

Since discovering the right biomarkers to target for a certain disease is a key challenge for medical research, the second potential of nanopharmacy in medical and pharmaceutical use is if nanoparticles are replaced with nanomaterials, whose symmetry is similar to biological water and biomolecules. Bearing in mind that the energy states (electronic, vibrational, rotational, and translational) of materials are defined by their symmetry, nanomaterials will have greater efficiency than nanoparticles in biomedicine. One example of nanomaterials with elements of icosahedral symmetry, such as biological water, and some biomolecules such as collagen, microtubules, and clathrin, is the C_60_ fullerene molecule. To overcome the problems of solubility of C_60_ molecules in water and its toxicity, the first derivative (FD-C_60_ or fullerenol) was created in the 90s of the last century. Molecule C_60_ is added with different numbers of OH groups by covalent bonds. Many studies of the use of FD-C_60_ have been carried out in biomedical research, and it has been shown that it has good effects at certain concentrations in photoinduced DNA, inhibits HIV-1 protease, generates specific interactions with proteins (fullerene-specific antibodies), has skin effects, nerve protection, antitumor activity, and others [4]. It has been shown that the toxicity of C_60_ is reduced by approximately 50% if OH groups are added [5]. Fullerene hydroxylation increases water solubility and affects how these nanomaterials interact with biological systems. It has been demonstrated that increasing fullerene water solubility through surface modification is related to significantly decreased toxicity. Specifically, this study observed decreased toxicity of hydroxylated fullerene compared to the cytotoxic effects of fullerene aggregates in human skin (HDP) and liver carcinoma (HepG2) cells [5]. Similarly, hydroxylation was observed to decrease the toxic potential of fullerene in mouse L929 fibrosarcoma, rat C6 glioma, and U251 human glioma cell lines [6]. Additionally, hydroxylated fullerene induced apoptotic changes in the investigated cell lines, while fullerene C_60_ induced necrotic cell death. The distinct effects of pristine and modified fullerene originate from the different nanoparticle interactions with the intracellular metabolic pathways [5,6]. In human breast cancer cell lines, C_60_(OH)_20-24_ inhibited cancer cell growth and suppressed doxorubicin-induced cytotoxicity [7]. Fullerenol C_60_(OH)_20-24_ (FD-C_60_) showed antitumor and anti-metastatic activity in vitro and in vivo EMT-6 breast cancer metastasis model [8]. The anti-tumor effect of fullerenol C_60_(OH)_20_ may be exerted through its effects on oxidative stress status, inhibition of the formation of angiogenesis factors, or through modulation of the immune profile [9]. However, fullerene hydroxylation did not provide the absolute absence of toxicity in living systems [10,11].

In both cases, the first and second potentials of nanopharmacy, biological water were the environments in which nanopharmaceutical substances (nanoparticles and nanomaterials) acted. The first derivative of C_60_ (FD-C_60_ or fullerenol) belongs to the group of biochemical substances, while our approach to the second and third derivatives of C_60_ (SD-C_60_ and TD-C_60_) belongs to the group of biophysical substances. The basic differences between the biochemical (self-assembly) and biophysical (self-organization) properties of C_60_ derivatives are given in the discussion. In our approach, based on stable water “helix chains” and water bubbles, “micelles” the third potential of nanopharmacy in biomedicine is that water is not only an environment in which biochemical processes take place but is also an active factor in biophysical processes that are realized through covalent and non-covalent hydrogen bond oscillations. In our research, we applied biomimicry, the principles of symmetry and harmony, to the functioning of water, DNA, and proteins based on the oscillatory processes of their hydrogen bonds. As non-covalent hydrogen bonds have classical and quantum properties [12], and as the C_60_ molecule has a *Re* (real) and *Im* (imaginary) electromagnetic spectrum [13], a method was developed to obtain the second derivative of the C_60_ molecule (SD-C_60_ or 3HFWC) as a result of the effect of the *Re* spectrum of the external electromagnetic oscillatory field and *Re* spectra of C_60_ on water around the FD-C_60_. In such a process, stable water layers are formed around FD-C_60_ [14,15,16,17]. However, when there is the effect of the *Re* and *Im* spectra of the external electromagnetic oscillatory field and the effect of the *Re* and *Im* spectra of C_60_ on the FD-C_60_ and water in the reactor, stable water layers are formed around FD-C_60_ (SD-C_60_ or 3HFWC), and for the first time stable water “helix chains” and water bubbles (“micelles”) are formed in the reactor (TD-C_60_ or 3HFWC-W). This new approach, based on stable water structures (layers, chains, and water bubbles), to nanopharmacy (SD-C_60_ and TD-C_60_) opens up the possibility of a new method for personalized medicine because it accesses the functionality of DNA and proteins through a biophysical energy-informational approach of hydrogen bonds that is specific for each individual.

The effects of the third potential of nanopharmacy (SD-C_60_ and TD-C_60_) in biomedical research on the treatment of skin cancer, Alzheimer’s, pain, and memory in mice are presented based on this paper. However, this third approach to nanopharmacy (SD-C_60_ and TD-C_60_) opens up the possibility of a new approach to personal medicine because it accesses the functionality of DNA and proteins through a biophysical energy-informational approach of hydrogen bonds that is specific for each individual. Nano-physical pharmacy based on hydrogen bonds of DNA, proteins, and water may play a pivotal role in advancing personalized medicine by providing innovative tools and platforms for precise diagnosis, targeted therapy, and monitoring of treatment responses.

For the first time, we have shown that the synthesis of stable water “helical chains” and water bubbles, “micelles”, is possible under the influence of an external oscillatory magnetic field, when the precursor is FD-C_60_. The substances SD-C_60_ and TD-C_60_ significantly increased the solubility of fullerene derivatives in water and reduced their toxicity. This enables wider biomedical research, and initial studies show promising applications in cancer, Alzheimer’s and other diseases.

## 2. Materials and Methods

### 2.1. Sample Preparation

The two basic materials used in this experiment are water and the fullerenol C60(OH)36) or FD-C60. The C_60_ molecule is composed of 60 carbon atoms that are organized in 12 pentagons and 20 hexagons on the surface of the sphere.

Tap water was treated using reverse osmosis, and finally the composition of the water in the experiment was Ca^2+^—0.60 mg/L, Mg^2+^—0.53 mg/L, Na^+^—0.32 mg/L, K^+^—0.04 mg/L, Fe^2+/3+^—<0.005 mg/L, NH_4_^+^—<0.03 mg/L, Cl—0.18 mg/L, and NO_3_—0.12 mg/L, with conductivity in the tank of 0.05 mS/cm. Water with this characteristic was used for the production of both SD-C_60_ and TD-C_60_. More information about water and its hydrogen bond properties can be found in papers [16,17] and Appendix A. As a precursor of SD-C60, the fullerenol C60(OH)36 or FD-C60, with a molecular weight of 1332 Da, a dust composition (yellow color), and a purity of 99.99%, was ordered in a dark bottle from Solaris Chem, Vaudreuil-Dorion, QC, Canada. It was stored until used in a dark room, with a humidity of 35 ± 2% and a temperature of 20 ± 2 °C.

The C60 derivatives were synthesized at the NanoWorld Lab (TD-C_60_) and TFT Nano Center (SD-C_60_), Belgrade, Serbia (3 g of fullerol was mixed with 20 L of ultra-pure water), according to the patented procedure [14,15]. The formation of C60 derivate began with 0.150 g/L of C60(OH)36 dissolved in high-purity water (0.05 μS/cm) under the influence of an external oscillatory magnetic field of +250/−92 mT [H(*ωt*) = *Acos*(ωt) + *Bisin* (ωt), ω = *f* (*R*,*L*,*C*)] according to the icosahedral eigenvalues T1u, T_1g_, T2u, T_2g_ (Fibonacci numbers ±Φ and, ±ϕ), Appendix A. At the same time, under the internal action of the vibrational energies of the C60 molecules (same vibration law as an external *Re* and *Im* magnetic field) in a reactor at 37 °C, the formation of a few C_60_ derivatives was realized (Figure 1, Appendix A).

### 2.2. Sample Characterization

Investigations of C60 derivatives were characterized using UV-VIS-NIR, FTIR TEM/STEM, and AFM/MFM techniques.

#### 2.2.1. UV–Vis–NIR and FTIR

UV–Vis–NIR characterization of C_60_ derivatives was performed using a Lambda 500 spectrometer, Perkin-Elmer, Waltham, MA, USA, in the range of 250–3000 nm. FTIR characterization was performed using the Spectrum Spotlight 400 FTIR Imaging System, Perkin-Elmer, Waltham, MA, USA, in the range of 2500–16,000 nm (the image can be selected and its spectra can be obtained).

#### 2.2.2. TEM and HAADF-STEM

TEM (transmission electron microscopy) analysis of the dry particles of C_60_ derivatives was performed in order to determine their solid-state size and shape. The samples in a liquid state were applied to the TEM copper mesh coated with carbon and dried in air. After drying the samples, they were analyzed and recorded in three TEM laboratories: (1) the CM12 Philips/FEI Transmission Electron Microscopy, Eindhoven, The Netherlands, magnification ×45,000 and ×60,000; (2) TEM, JEM 1400, JEOL, Tokyo, Japan, magnification ×120,000 up to ×200,000; and (3) High-Angle Annular Dark Field Scanning Transmission Electron Microscopy (HAADF-STEM), Thermo-Fisher Talos/Osiris 200 kV (Waltham, MA, USA), Zeiss Libra (Oberkochen, Germany) 120 kV, 200 kV electron energy, STEM mode with bright field, with evaluation software Thermo-Fisher Velox 3.7 and energy-dispersive X-ray analysis EDXS mapping = ChemiSTEM, with Z > 8 installed.

#### 2.2.3. AFM/MFM

Samples of C_60_ derivatives were characterized using JSPM-5200, Scanning Probe Microscopy, JEOL, Akishima, Tokyo, Japan. Two methods were used: (1) AFM (Atomic Force Microscopy), and (2) MFM (Magnetic Force Microscopy). Both techniques are non-invasive. The AFM method is based on Van der Waal’s forces and London-type dispersive forces between the tip and the sample, while MFM, in the non-contact imaging mode, is based on magnetic dipole-dipole interaction (hydrogen bonds) between the tip and the sample (measuring deflection of tip “ϕ” in deg.). For magnetic gradient investigation, specialized cantilevers, type HQ NSC18/Co-CrAl BS (MikroMasch, Tallinn, Estonia), with force constants in the range between 1.2 and 5.5 N/m and with the resonant frequency range between 60 and 90 kHz, were used. The scanning size of the sample depended on the object size and the number of objects that should be scanned. In this case, the optimal scan size should be between 10 nm and 100 nm.

### 2.3. Methods

The method is based on three main elements that are present in the laws of nature: *symmetry*, *harmony*, and *perfection*. All the laws of nature that we know have symmetry at their root, the processes that take place are optimized (harmonized), and embryogenesis in biological systems is one of the best examples of perfection (from one fertilized cell, a precisely determined organism with several billion cells is created).

#### 2.3.1. Symmetry

The symmetry of the structure will determine its energy states (electronic, vibrational, and rotational). Since C_60_ is a molecular crystal with icosahedral symmetry, it will transmit its vibrational effects through hydrogen bonds into the surrounding water space. When water arranges itself into water chains or clusters, it also does so according to the laws of symmetry. Biomolecules with hydrogen bonds in their structure, such as DNA, collagen, microtubules, clathrin, and other proteins, are complex symmetrical objects. However, they all have the same electronic and vibrational states of covalent (molecular, O-H, and N-H) and non-covalent (intermolecular O…H and N…H) hydrogen bonds. Because of this, it is possible to transmit signals from one to another to a third object. In our case, it is the transmission of vibrational signals from C_60_ molecules through hydrogen bonds of water to biomolecules such as DNA, collagen, microtubules, clathrin, etc.

#### 2.3.2. Harmony

The key element for achieving harmony in hydrogen bonds, and thus also achieving the optimal conformational state of biomolecules, is the ratio of covalent (O-H or N-H) and non-covalent (O…H or N…O) hydrogen bonds. Experimentally, using neutron diffraction, it was determined that the value of O-H vs. O…H is within the limits of 1.2 to 2.1 [18]. However, the optimal (harmonic) value is approximately 1.62 [16]. In order for the conformational states of biomolecules to be optimal, that is, for biological water to perform an optimal function (good interaction with biomolecules), the value of O-H/O…H should be approximately 1.618…(icosahedral symmetry element Φ = 1/2(1 + $\sqrt{5}$), which is related to the pentagonal structure of the C_60_ molecule and water molecules).

#### 2.3.3. Perfection

This third element of the method, perfection, which is based on the laws of perfect numbers, is present in the molecule C_60_ because this molecule has 20 hexagons (the number 6 is the first perfect number because the sum of its divisors 1, 2, and 3 makes itself). This element of the method is very significant if the effect of C_60_ molecule derivatives on the embryogenesis is taken into account. The reason for this lies in the fact that the creation of an organism from a fertilized egg cell (embryogenesis) to its arrival in the world takes place according to the laws of body symmetry, the harmony of the formation and organization of cells in tissues and organs, and the creation of the body’s biophysical information networks (quantum-classical) and codes according to the laws of perfect numbers [19].

Figure 2 schematically represents the SHP (symmetry-harmony-perfection) method and its connection with the structures considered in this work (derivatives of C_60_ molecules, DNA, proteins, and water) from the aspect of hydrogen bonds.

The importance of hydrogen bonds for the functioning of the organism was first noticed in 1939 by Linus Pauling, who said, “I believe that as the methods of structural chemistry are further applied to physiological problems, it will be found that the significance of the hydrogen bonds for physiology is greater than that of any other single structural feature”. Pauling’s work on understanding the importance of hydrogen bonds in chemistry and biology, based on experimental results and the theory of quantum mechanics, helped demonstrate its importance in determining the properties of substances and their interactions [14].

## 3. Results

### 3.1. FD-C_60_ (The First Derivative of the C_60_)

In the experiment, five different structures of C_60_ derivatives are obtained, which are shown in Figure 3, Figure 4, Figure 5 and Figure 6. Figure 3 shows the first derivative of the C_60_ molecule (FD-C_60_), size 1.2 nm, in the form of a schematic representation (a), TEM image (b), and in solution (c) (0.15 g/L dissolved in water is medium brown in color). Physicochemical characteristics of FD-C_60_ are presented in reference [17].

### 3.2. TEM Images of SD-C_60_ and TD-C_60_ Derivatives of C_60_

There are three types of SD-C_60_, depending on the type of oscillatory magnetic field used to form water layers around FD-C_60_ (as a precursor) and water chains that are formed in the water solution of the reactor. If an oscillating magnetic field of the type H(ωt) = *cos*(ωt) + *isin*(ωt) is applied, then we get the first type of the second derivative C_60_: SD-C_60_, which we marked in previous works as 3HFWC, which we will now mark as SD-C_60_-(*f*), where *f* = ω/2π (Figure 4). However, if we apply an oscillatory magnetic field of the type H(ωt) = *cos*(ωt), then due to the nature of the field in the frequency domain, the two solutions are *f*_1_ = 1/2 *e^iωt^* (positive frequency) and −*f*_1_ = 1/2 *e^−iωt^* (negative frequency). Two structures of the second derivative C_60_ are formed: SD-C_60_-(*f*_1_) and SD-C_60_-(−*f*_1_), which are mixed and which in previous works we marked 3HFWC-W. Notation (*f*_1_) and (−*f*_1_) means that the structures originate from the *Re* (real) spectrum of the oscillatory magnetic fields (C_60_ and reactor). Until now, only the structure SD-C_60_-(*f*_1_) has been observed, which we identified with SD-C_60_-(*f*) because they are the same structures in character, only differing in the number of water layers (Figure 5a). However, if an oscillatory magnetic field of the type H(ωt) = *isin*(ωt) is applied, then due to the character of the field, a third derivative C_60_ (TD-C_60_) is formed with two water structures, TD-C_60_-(*f*_1_^∗^) and TD-C_60_-(−*f*_1_^∗^) (Figure 6). They are the result of the composition of the *Im* (imaginary) spectrum of the oscillatory magnetic field of the reactor and the imaginary spectrum C_60_.

### 3.3. FTIR Spectra of SD-C_60_ and TD-C_60_ Derivatives of C_60_

The FTIR spectrum, *Re*, and *Im*, of the C_60_ molecule are well-known [13] (Appendix A). The FTIR spectra of FD-C_60_ and SD-C_60_-(f) were also performed, and the diagrams show significant differences in the 3000–3300 nm domain where hydrogen bonds are dominant [16]. In this paper, we have presented the spectrum of SD-C_60_-(f), Figure 7, [17] so that it could be compared with the performed FTIR spectra for SD-C_60_-(*f*_1_,−*f*_1_) (Figure 8) and TD-C_60_-(*f*_1_^x^,−*f*_1_^x^) (Figure 9).

As can be seen from the diagram (Figure 7, Figure 8 and Figure 9), there are great similarities in the spectra as well as small differences. This indicates that the structures are of the same symmetry in all three cases. In other words, aqueous layers where the C_60_ molecule is physically present and aqueous layers where the C_60_ molecule is not physically present oscillate approximately the same. The position of the peaks is approximately the same, but the differences in intensities and peak widths are different. The main peak at approximately 3000 nm (hydrogen bonds) is present in all three cases, which means that these structures are composed of water. A maximum wavelength shift between these structures is 66 nm. The intensities are three times different in favor of the third derivative compared to the second derivative. The second peak at approximately 4700 nm is present in all three cases with a very small wavelength shift of 20 nm, with the intensity of the peak in the third derivative being more than 10 times higher than in the second derivative. The third peak at approximately 6100 nm is present in all three cases with a wavelength shift of approximately 160 nm, with the intensity of the peak in the third derivative being more than three times higher than in the second derivative.

### 3.4. MFM Spectra of SD-C_60_ and TD-C_60_ Derivative of C_60_

MFM spectra show hydrogen bonds in all five structures of C_60_ derivatives are present (Figure 10). MFM precisely determines the intensity of hydrogen bond dynamics. The intensity of paramagnetic spectra is very high, meaning that C_60_ derivatives are very rich in water molecules because they interact with the MFM tip (dipole-dipole interaction). The results prove the water molecules’ presence and the intensity of water dynamics in all five structures of C_60_ derivatives. Blue lines (Figure 10) show moisture presence in the precursor. The intensity of hydrogen bond dynamics in the C_60_ derivatives is many times greater than in the case when there is only present moisture in FD-C_60_.

### 3.5. “Yin-Yang” Machinery of C_60_ Molecule Derivatives Synthesis

The C_60_ molecule is both a classical and a quantum entity because it behaves as a particle and a wave [20]. It transfers its symmetry-harmony-perfect physical properties to the water layers around it, consisting of water molecules whose hydrogen bonds are also classical-quantum [12]. Through vibrational modes Φ,−ϕ,−Φ,ϕ its harmonized vibrational states are transferred to water, and it further transfers vibrations to biomolecules that have hydrogen bonds (DNA and proteins, such as collagen, microtubules, clathrin, actin, etc.).

The oscillatory magnetic field equation H (ωt) = A *cos* (ωt) + B *isin*(ωt) gives the possibility to make three types of derivatives of C_60_ molecules: ❶, ❷ and ❸ (Figure 11). The first form (❶) of the second derivative of the C_60_ molecule (SD-C_60_) is created under the influence of the magnetic oscillatory field of the reactor H(ωt) = *A*_0_* e^i^*^πω^ (3HFWC or *f*) and the vibration of the C_60_ molecule. The second form (❷) of the second derivative of the C_60_ molecule (SD-C_60_ (3HFWC-W) or *f*_1_,−*f*_1_), which has two structures (*f*_1_ and −*f*_1_), is formed under the effect of the *Re* magnetic oscillatory field of the reactor H(ωt) = *Acos*(ωt), and the vibration of C_60_ molecules. The third form (❸), as a third derivative of C_60_ molecules (TD-C_60_ or *f*_1_^∗^,−*f*_1_^∗^), has also two structures (*f*_1_^∗^ and −*f*_1_^∗^). It is created under the effect of the *Im* magnetic oscillatory field of the reactor H(ωt) = *B isin*, and the vibration of the C_60_ molecules.

In the system of icosahedral symmetry, to which C_60_ belongs, not only two paired eigenvalues Φ,−ϕ are harmonic, but also the system of two paired values (square:Φ,−ϕ,−Φ,ϕ) is harmonic. Because of such relationships, the system can be represented as the ancient Chinese *Yin-Yang* system: big *Yin* (−Φ), small *yin*(−ϕ), big *Yang* (Φ), and small *yang* (ϕ) (Figure 12 left). The synthesis of TD-C60 is a unique process in which a substance is obtained with four components that are harmonized by their frequency characteristics to act harmoniously on hydrogen bonds of water and biomolecules (Figure 11).

The vibrational modes of the oscillatory magnetic field of the reactor are also matched according to the laws of symmetry and harmony. The elements of Euler’s formula [*e*^iωπ^ = *cos*(ωt) + *isin*(ωt)] also form a harmonic logical square (1/2*e^iωt^*, 1/2*e*^−*iωt*^, *e*^−*iωt*^/2*i*, −*e*^−*iωt*^/2*i*), as it is the case with the ancient Yin-Yang (Figure 12, right).

These two “Yin-Yang” systems (based on *e*^iωπ^ and −Φ, ϕ, Φ, −ϕ) are the “heads and tails” of a coin based on the symmetry-harmony-perfection of the C_60_ molecule and the oscillatory magnetic field of the reactor that creates derivatives.

The presented “Yin-Yang” systems (Figure 12) are not given symbolically but are real physical (*Re*, *Im*) properties of C_60_ molecules and their derivatives. In the case of Figure 12 right, there is an interference of two electromagnetic waves (C_60_—inner and reactor-external) that interact and maintain a harmonized relationship (Yin-Yang) of their amplitudes in that interaction. There is a similar case in the biphoton experiment when the amplitude and phase structure make the image that can be detected as Yin-Yang order [22] (Appendix A).

## 4. Discussion

The three most important things are discussed in this section. First, what are the similarities and differences between chemical nano-machinery and physical nano-machinery, secondly, what effects of C_60_ derivatives have been achieved in biomedical research, and thirdly, what would be the further direction of research.

### 4.1. Physical Nano-Machinery vs. Chemical Nano-Machinery

Similarities and differences between chemical and physical nano-machineries are given in Figure 13. Chemical approaches are based on covalent bonds, while physical ones are based on non-covalent bonds. Chemical is a self-assembly process, while physical is a self-organizing process. In chemistry, enthalpy (ΔH) is important, while in physics, next to enthalpy, entropy (ΔS) is also important. Chemical entities (molecules, like C_60_ and C_60_(OH)_x_) are stable with small variations in conformational changes, but physical layers of water molecules based on intermolecular interactions are prone to change. In the fact that in a given environment, water molecules return to a stable state because water, as the solvent, is an active factor in the process (primary solvent influence). As Lao Tzu said, “Nothing is softer, or more flexible than water, yet nothing can resist it”.

### 4.2. Effects of C_60_ Molecule Derivatives in Biomedical Applications

Polyhydroxylated fullerenes (fullerenols) or FD-C_60_ have attracted great interest in recent years as promising candidates for cancer therapy due to their exceptional properties, such as high water solubility, biocompatibility, biodegradability, and rapid elimination from the body. These nanomaterials are particularly valuable for cancer treatment because they can inhibit tumor growth and enhance the immune response against cancer cells [25]. Fullerenols (FD-C_60_) are also known for their ability to act as photosensitizers (PS), i.e., molecules that can be activated by specific wavelengths of light, providing an additional option for cancer treatment through phototherapy [26,27]. Although fullerenols are generally known for their antioxidant properties [28,29], they can act as prooxidants upon photoexcitation due to the generation of ROS [30,31], which makes them potential candidates for cancer phototherapy.

Our recent studies have demonstrated the potential of the second fullerene derivative, 3HFWC or SD-C_60_ as an antitumor agent in melanoma in vitro and in vivo models [32,33] in combination with light treatment. Specifically, cells exposed to 3HFWC (SD-C_60_) were irradiated with incoherent, phase-shifted, unsynchronized, polychromatic, hyperpolarized light (HPL) (Bioptron-2 device, Bioptron AG, Wollerau, Switzerland) [34]. The nanophotonic fullerene-filtering light of this device allowed the emission of a broad spectrum of wavelengths (light from 400 to 1100 nm, and infrared (IR) irradiation from 5000 to 15,000 nm, with characteristic peaks at 5811 nm, 8732 nm, and 13,300 nm). In vitro experiments performed with both primary cells and melanoma cell lines of different grades and invasiveness (low-grade B16 and high-grade B16-F10 mouse cells and human A375 cells), demonstrated the selective anticancer effect of 3HFWC. This effect was attributed to the induction of senescence and/or differentiation towards the melanocytic phenotype. Electron microscopy analyses also showed that 3HFWC was efficiently internalized by the cells, indicating this process is an important step in the reprogramming of cancer cells [32]. Subsequently, the anticancer effect of 3HFWC (SD-C_60_) was confirmed in vivo using a syngeneic mouse melanoma model. Mice were treated with 3HFWC via drinking water (0.145 mg/mL, ad libitum) after tumor induction alone or in combination with HPL irradiation of animals (2 × 20 min daily). Again, 3HFWC-induced senescence and differentiation of melanoma cells were observed. Senescent cells were larger with enlarged nuclei, more lipid droplets, increased lysosomal activity, and altered mitochondrial morphology. On the other hand, cells that had undergone melanocytic differentiation had smaller, heterochromatic nuclei, increased melanin production, and well-developed dendrites with mature melanosomes [32,33]. Cell senescence is a stable state characterized by cell cycle arrest, and is considered as a response to cell damage caused by various stressors [35,36]. This process is considered beneficial in cancer therapy as it reduces the malignant potential of neoplastic cells in contrast to traditional cytotoxic approaches, such as chemotherapy and radiotherapy [37,38,39]. Inducing senescence in cancer cells may be beneficial as it reduces their aggressiveness and regulates the tumor microenvironment (TME). Senescent cells are known to secrete proinflammatory cytokines and matrix metalloproteinases [40,41], which affect the cancer cells and other cells and extracellular matrix components within the TME [42,43]. Senescent cancer cells can recruit immune cells to the site, and thus promote antitumor immune surveillance [44,45,46,47,48]. In melanoma models, treatment with 3HFWC (SD-C_60_), especially in combination with HPL irradiation, stimulated the infiltration of tumor-suppressive immune cells, such as CD8+ cytotoxic T lymphocytes. In contrast, the presence of tumor-promoting immune cells, such as T regulatory cells, myeloid-derived suppressor cells, and M2 macrophages was reduced in TME [33]. These results suggest that 3HFWC also has immunomodulatory effects and thus converts “cold” (non-inflamed) tumors into “warm” (inflamed) tumors that might respond better to immunotherapy [49,50,51,52].

In summary, the approach to the development of different types of micro/nano-systems and devices, as we have shown in this paper (a combination of 3HFWC substance and HPL light during the treatment), increases the potential and efficiency of micro/nano-system applications in biomedicine. The combination of 3HFWC treatment and HPL irradiation could represent a novel approach to cancer therapy, not only by directly reprogramming and destroying tumor cells, but also by activating an immune response that attacks and eliminates the remaining neoplastic cells.

This dual mechanism of action–cellular reprogramming and activation of the immune response–distinguishes it from conventional cytotoxic treatments. Although HPL irradiation enhanced the effect of 3HFWC, it did not induce the same strong oxidative stress typically associated with PDT. This suggests that the combined treatment works via a different mechanism, possibly through the organization of water dipoles around the 3HFWC molecules, which enhances their anti-cancer effect without causing significant cytotoxicity [32].

Further studies on the prophylactic use of 3HFWC (SD-C_60_) as well as its combination with other treatment modalities are needed to fully understand its potential as a cancer therapy. Initial results suggest that administration of 3HFWC (SD-C_60_) after tumor induction is more effective than preventive treatment, which could impair the early antitumor immune response [53,54]. Toxicity studies in normal cells did not show severe systemic toxicity. However, mild liver and kidney damage was observed in some cases, which warrants further investigation to determine the optimal dosage and treatment duration before potential use in humans [33].

Compared to the previously described 3HFWC (SD-C_60_-f, Figure 4) compound, the improved formulation labeled 3HFWC-W (SD-C_60_-*f*_1_,−*f*_1_), (Figure 5) showed a significantly amplified potential to limit melanoma cell growth in vitro. Despite this enhanced activity, the mechanism of action remained unchanged, as it primarily induced cellular senescence and did not lead to cell death [16].

The potential application of fullerene C_60_ and its modifications in the tertiary stages of Alzheimer’s and Parkinson’s disease is given in a review paper [55]. Our research group has shown that SD-C_60_ (3HFWC) has effects on the mechanisms that lead to Alzheimer’s disease because it prevents 30% of the formation of new plaques in patients [56,57].

The synthesis of SD-C_60_-*f* (3HFWC-W) with a magnetic field of the type H(ωt) = A*cos*(ωt) is similar to stepwise oscillatory circuits of DNA molecules with current law *I* = 5.2*e*^−0.005t^*cos*(0.068t), where A = 5.2*e*^−0.005t^ and the angular velocity of the oscillation ω = 0.068 [58]. The under-damped oscillation with parameters *C* = 0.02pF, *L* = 0.01H, and *R* = 100 has a frequency value, according to equation ω = $\sqrt{\frac{1}{LC}-\frac{R^{2}}{{4L}^{2}}\;}$, is approximately 10^10^ s^−1^ (Appendix A). It is a similar value to the ^13^C-NMR experimental result of C_60_ a rotational diffusion constant D = 1.8 × 10^10^ per second [59].

### 4.3. The Further Direction of Research

Bearing in mind what was said in Section 3.4 and Section 4.2, it is necessary: (1) to continue with the chemical-physical characterization of the second (SD-C_60_-*f*_1_,−*f*_1_) [3HFWC-W] and third derivative (TD-C_60_) of the C_60_ molecule, (2) to examine the biomedical effects of the second derivative C_60_/(*f*_1_,−*f*_1_) and the third derivative C_60_/(*f*_1_^∗^,−*f*_1_^∗^), separately, and (3) mixture of them as a Quantum-classical substance (QCS) for medicine, which is a synergy of the second derivative SD-C_60_/(*f*_1_,−*f*_1_) and the third C_60_(*f*_1_^∗^,−*f*_1_^∗^) derivatives (QCS = (*f*_1_,−*f*_1_)/(*f*_1_^∗^,−*f*_1_^∗^ = “Yin-Yang” substance, Figure 12, right).

Derivatives of the C_60_ molecule, second and third, can be used in biomedical research individually, but the best result will be obtained if they are used as a complex whose percentage representation is obtained during production (ϕ + ϕ^2^ = 1) so that the percentage of time of 38% is for the production of SD-C_60_ (A*cos* (ωt) and 62% for TD-C_60_ production (B*isin*(ωt).

## 5. Conclusions

The task of science in general, and biomedical engineering in particular, is to review existing knowledge and technological solutions and to propose, based on theoretical and experimental knowledge, new methods, techniques, and products that contribute to the improvement of human health. The results presented in this paper support the fact that in the field of pharmacy, especially nanopharmacy, there is a possibility of improving human health by using not only chemical (FD-C_60_) but also physical methods (SD-C_60_ and TD-C_60_) to manufacture products. A particularly important area is hydrogen bonds, which are the most widespread interactions in biological systems because they are (approximately 85%) an integral part of the intermolecular interactions of water, which is 60–70% of the organism, DNA (A=T, C=G), and proteins (O…H and N…H), which make up approximately 16% of the structure of the human organism.

The paper presents the method of obtaining derivatives of C_60_ molecules and their characterization using FTIR, TEM, and AFM/MFM. It was shown that the size of the derivatives is approximately 15 nm and that they are rich in water molecules arranged in water layers with a high degree of vibrational coherence.

A brief overview of the initial biomedical research on the application of derivatives of C_60_ molecules indicates that these nano-physical substances have good effects in the treatment of cancer, Alzheimer’s, and other diseases. To further test the existing derivatives of the C_60_ molecule, it is necessary to conduct a biomedical study to determine the effects of the newly obtained an integral substance of C_60_ derivatives, *Quantum-classical substance (QCS)*, which is a harmonized unity (“Yin-Yang”) of the second C_60_/(*f*_1_,−*f*_1_) and the third C_60_(*f*_1_^∗^,−*f*_1_^∗^) derivatives.

## Figures and Tables

**Figure 1 micromachines-16-00770-f001:**
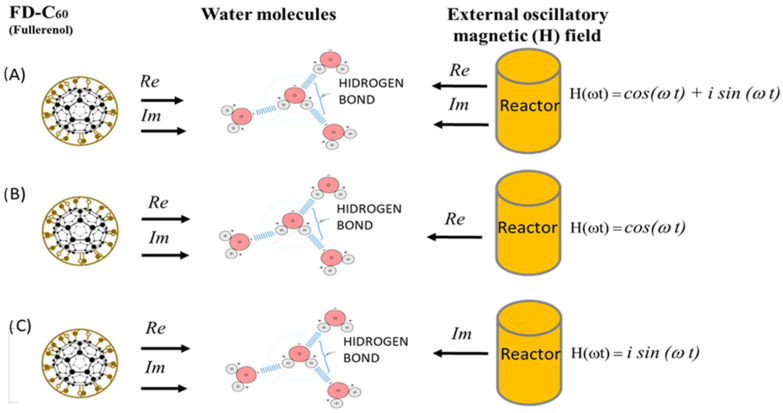
The method of production of C_60_ derivatives (SD-C_60_ and TD-C_60_) according to the principle “between a hammer and an anvil” (ref. [16]: Supp. Figure 5) in the frequency domain. In this process, water molecules are located between two electromagnetic fields of action: The C_60_ molecule and the external one (reactor). The C_60_ molecule has its natural permanent *Re* and *Im* electromagnetic effect (Appendix A) on the water molecules in its environment. The external effect (reactor) can be selective: (**A**) both *Re*, *Im*, (**B**) *Re*, and (**C**) *Im* part. Depending on which combinations of *Re* and *Im* are realized, there will be different organizations of water molecules in layers around FD-C_60_ and in water (chains and cluster/“micelles”).

**Figure 2 micromachines-16-00770-f002:**
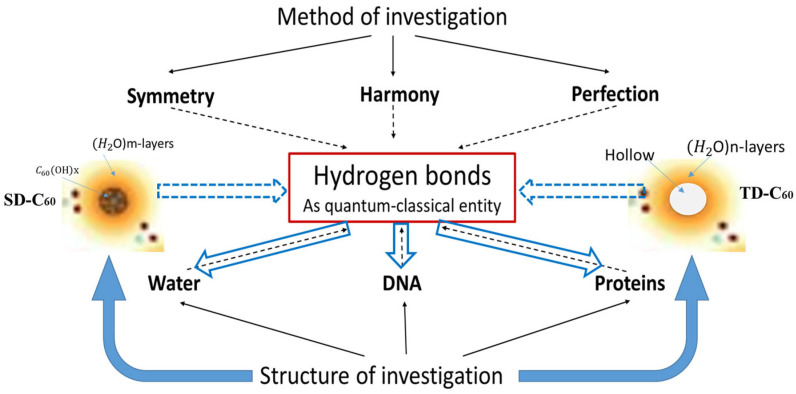
Schematic representation of the object of research (hydrogen bonds in DNA, proteins, and water) and the method of obtaining C_60_ derivatives (SD-C_60_ and TD-C_60_) with hydrogen bonds and their influence on DNA, proteins, and water. Solid and dashed black lines show functional dependencies between method and material, while blue arrows show the effect of newly created nano-materials (SD-C_60_ and TD-C_60_) on hydrogen bonds of DNA, proteins, and water. In this method, water is not only a medium in which processes take place, but its hydrogen bonds are an active participant in the process.

**Figure 3 micromachines-16-00770-f003:**
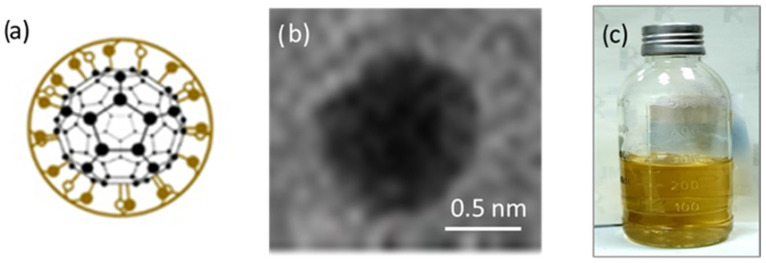
(**a**) FD-C_60_ (fullerenol) comprises molecule C_60_ and 36 OH groups attached by covalent bonds. (**b**) In a solid dry state, it is 1.2 nm in size, while under the influence of moisture, it is approximately 1.6–1.8 nm. (**c**) It dissolves well in water and, depending on the concentration, is dark to light brown.

**Figure 4 micromachines-16-00770-f004:**
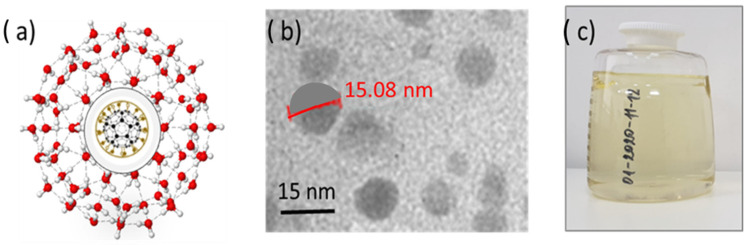
SD-C_60_-(f) [3HFWC]: (**a**) The structure consists of C_60_ molecules to which OH groups are connected by covalent bonds and water layers where the water molecules are arranged according to 3D Penrose tiling (3DPT) [16] (**b**) TEM Image of SD-C_60_-f in different sizes, from 9 to 16 nm, (**c**) the color of the derivative will depend on the percentage of success in creating water layers around the precursor (FD-C_60_, fullerenol). When comparing the colors of the products from Figure 3c and Figure 4c (which contain the same percentage of fullerenol), it can be seen that the product in Figure 4c is much lighter than in Figure 3c, which means that the greater part (approximately 85%) of the fullerol is enriched with water layers, i.e., that the transformation of FD-C_60_ into SD-C_60_-(f) was successfully completed.

**Figure 5 micromachines-16-00770-f005:**
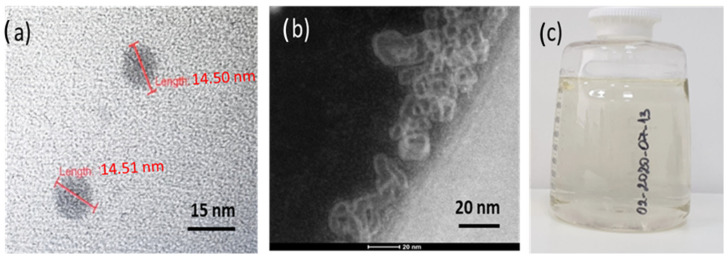
SD-C_60_-(f_1_,−f_1_) [3HFWC-W] is formed under the influence of the oscillatory magnetic field H(*ω*t) = *cos*(*ω*t). Two types of structures, (**a**,**b**) are formed and identified by TEM. Structure (**a**) is formed as a consequence of frequency *f*_1_ = 1/2e*^iωt^* and is similar to SD-C_60_-(f) (Figure 4a) because they originate from the *Re* spectrum of C_60_ and the reactor. The difference between *f* and *f*_1_ structures can be in the number of layers (size). The structure *f*_1_ is of the same size (approximately 15 nm), while *f*_1_ can be up to 30 nm in size. Water chain structure (**b**) is formed under the influence of frequency −*f*_1_ = 1/2e^−*iωt*^ (negative frequency of the *Re* spectrum of the reactor and C_60_). (**c**) Due to the significantly lower content of SD-C_60_-(*f*_1_) and the higher water content of SD-C_60_-(−*f*_1_), this complex (*f*_1_/−*f*_1_) has a much lighter brown color than structure *f* (Figure 4c).

**Figure 6 micromachines-16-00770-f006:**
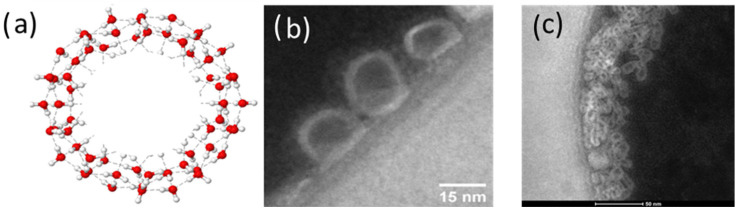
TD-C_60_-(*f*_1_^∗^,−*f*_1_^∗^) is formed under the influence of the oscillatory magnetic field H(*ω*t) = *isin*(*ω*t). Two types of water structures are formed. Structure (**a**,**b**) (a closed single “micelle” with a diameter of 15 nm and a shell thickness of 3–5 nm, arranged according to 3D Penrose tiling (radial direction) and pentagons and hexagons water molecule arrangement on sphere surface). This structure is formed as a consequence of frequencies *f*_1_^∗^ = *e^iωt^*/2*i*, while structuring (**c**) (a complex of closed cyclic water structures and open linear chain water structures) is formed under the influence of frequency −*f*_1_^∗^= −*e*^−*iωt*^/2*i.* Even if (b) and (c) do not contain the particle (C_60_ molecule), these structures belong to C_60_ derivatives because they were created under the influence of *Im* (imaginary) C_60_ spectra, and the reactor frequencies *f*_1_^∗^ = *e^iωt^*/2*i* and −*f*_1_^∗^ = −*e*^−*iωt*^/2*i*, which are also imaginary (*Im*).

**Figure 7 micromachines-16-00770-f007:**
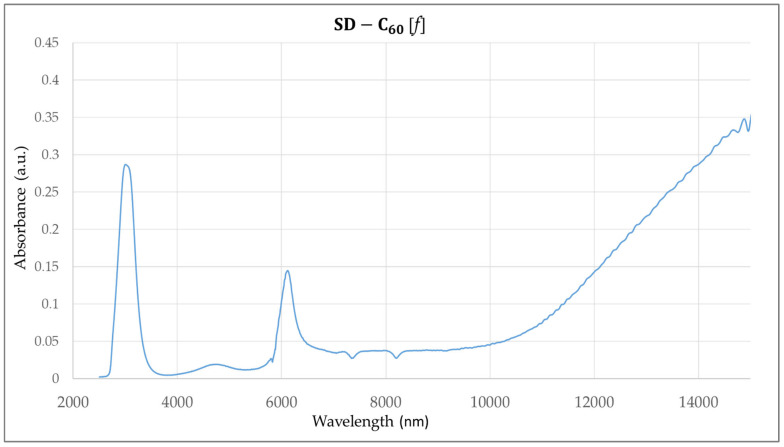
FTIR imaging spectra of SD-C_60_-(f) with four peaks with given wavelengths and intensities: [3044 nm, 0.2852], [4740 nm, 0.0192]. [6132 nm, 0.1431], [16,000 nm, 0.3500]. The peaks are not narrow, indicating that the structure’s oscillatory processes are not of a high degree of coherence.

**Figure 8 micromachines-16-00770-f008:**
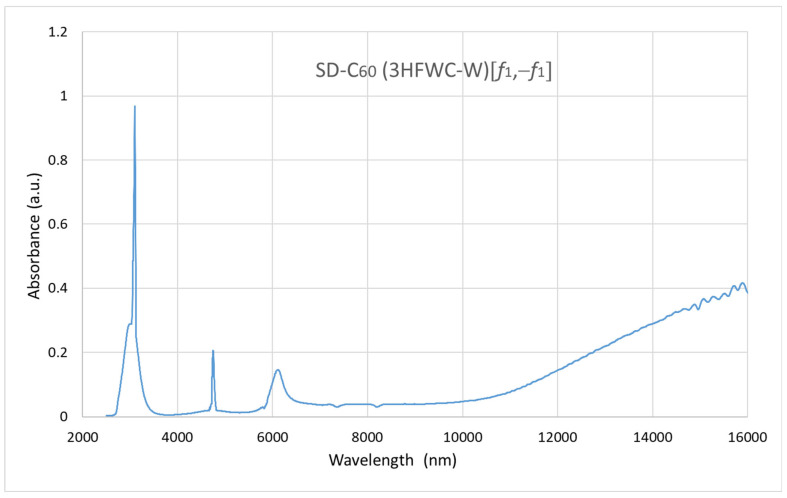
FTIR imaging spectra of SD-C_60_-(*f*_1_,−*f*_1_) with four peaks with given wavelengths and intensities: [3100 nm, 0.96325], [4748 nm, 0.2048], [6084 nm, 0.1433], [16,000 nm, 0.4000]. The first two peaks are narrow, which indicates a high degree of coherence.

**Figure 9 micromachines-16-00770-f009:**
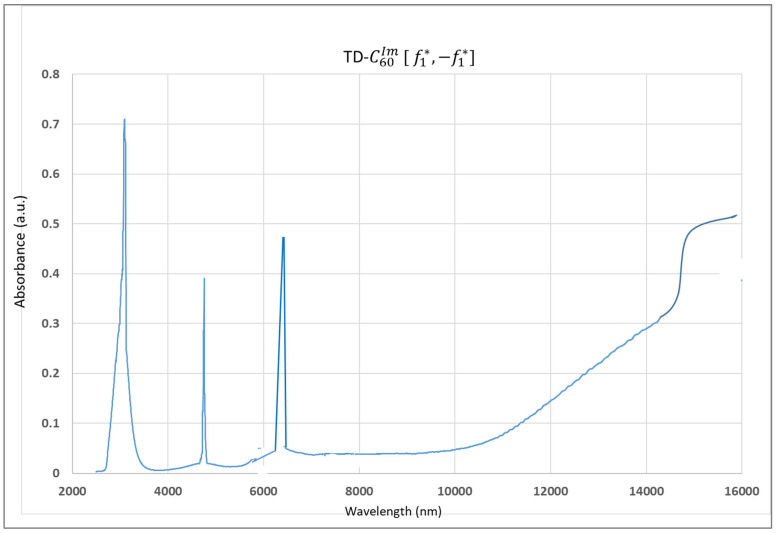
FTIR imaging spectra of TD-C_60_-(*f*_1_^x^,−*f*_1_^x^) with four peaks with given wavelengths and intensities: [3088 nm, 0.7086], [4760 nm, 0.3884], [6244 nm, 0.4813], [16,000 nm, 0.5162]. The first three peaks are very narrow, which indicates a high degree of coherence.

**Figure 10 micromachines-16-00770-f010:**
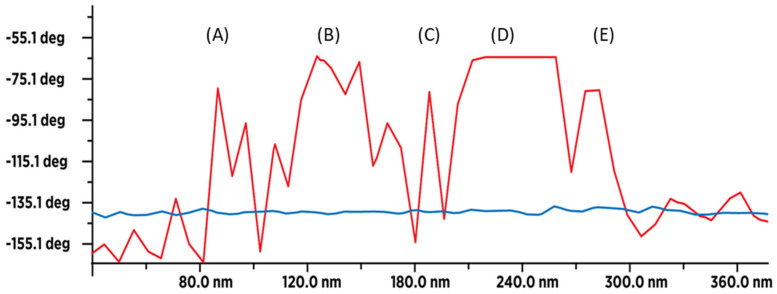
Magnetic Force Microscopy (MFM) diagram (scan size 360 nm) of the presence of hydrogen bonds in C_60_ derivatives (red line), while the blue line is the presence of moisture of the first derivative FD-C_60_ (fullerenol). As can be seen from the diagram, there are five basic structures of C_60_ derivatives: (A) SD-C_60_ (3HFWC-[*f*]), (B) SD-C_60_ (3HFWC-W[−*f*_1_]), (C) SD-C_60_ (3HFWC-W[*f*_1_] (similar structure as *f*), (D) TD-C_60_ ([*f*_1_^∗^]), and (E) TD-C_60_([−*f*_1_^∗^] (structure notation of *f*_1_,−*f*_1_, *f*_1_^∗^,−*f*_1_^∗^ is given on images, Figure 11).

**Figure 11 micromachines-16-00770-f011:**
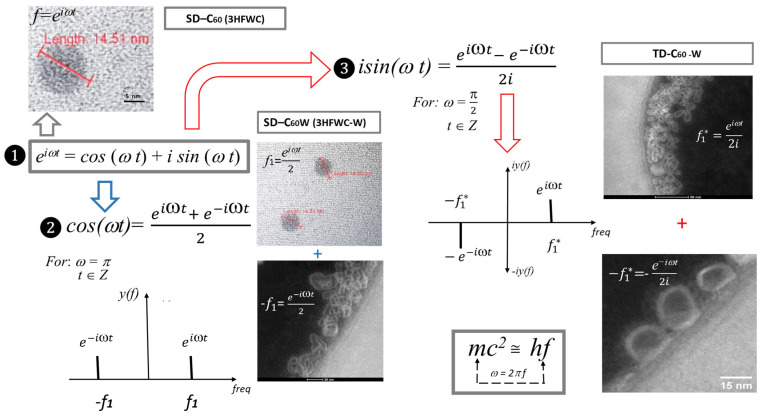
Summary schematic representation of SD-C_60_ and TD-C_60_ derivatives of C_60_ molecules. According to the frequency domain, the second derivative SD-C_60_ (❶) has two forms SD-C_60_-(*f*) or 3HFWC and SD-C_60_-(*f*_1_,−*f*_1_) or 3HFWC-W. The first form is formed as an interaction of the Φ vibration molecule C_60_ and the frequency (*f*) of the oscillatory magnetic field of the reactor. The second form of the derivative (❷) has two structures that are the result of the effect of the vibrational modes Φ and −ϕ of the C_60_ molecules on the water molecules in the reactor and the frequency of the *Re* oscillating magnetic field of the reactor *f*_1_ and −*f*_1_ [frequency diagram in the picture *y*(*f*)]. The third derivative (❸) also has two harmonized structures that arise as a result of the effect of the vibrations of C_60_ molecules (−Φ, ϕ) and the frequencies of the *Im* oscillatory magnetic field of reactors *f*_1_^∗^ and−*f*_1_^∗^ [frequency diagram in the picture *iy*(*f*)].

**Figure 12 micromachines-16-00770-f012:**
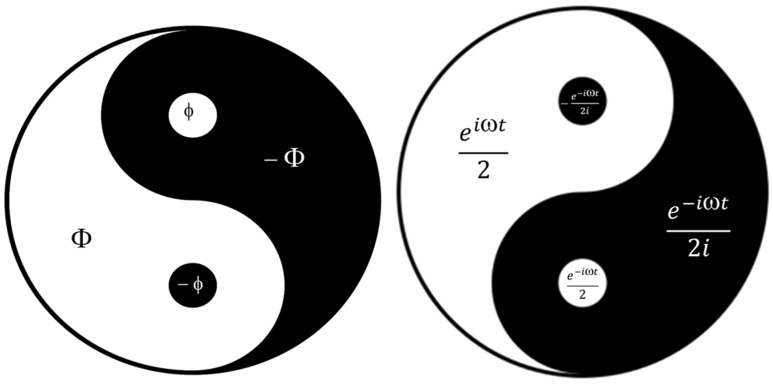
“Yin-Yang” machinery is a “head” and “tail” of symmetry/harmony of C_60_ molecule derivatives. The icosahedral symmetry of C_60_ contains four symmetry elements, Φ,−ϕ,−Φ,ϕ which ensure the harmony of structural-energetic stability. Its harmony transfers to the surrounding water molecules via hydrogen bonds. Simultaneously, the process (reactor) generates four frequency modes, *f*_1_, −*f_1_*, *f_1_*^∗^, −*f_1_^∗^* (which generate water structures according to the same law as C_60_). This is Euler–Fibonacci–Kostić (EFK) principle, because *f*_1_,−*f_1_*, *f_1_*^∗^, −*f_1_^∗^* are elements of Euler’s formula $e^{i\omega t}$ = *cos*(*ωt*) + *i sin* (*ωt*) and elements Φ,−ϕ,−Φ,ϕ are Fibonacci’s pairs. At the same time, their unity (“head” and “tails”) is Kostić’s coin solution, who said, “harmony is the synthesis of symmetry; symmetry is the analysis of harmony” [21]. This is exactly what happens to the structure and energy of the hydrogen bonds according to the Penrose process of 3D tiling formation [16].

**Figure 13 micromachines-16-00770-f013:**
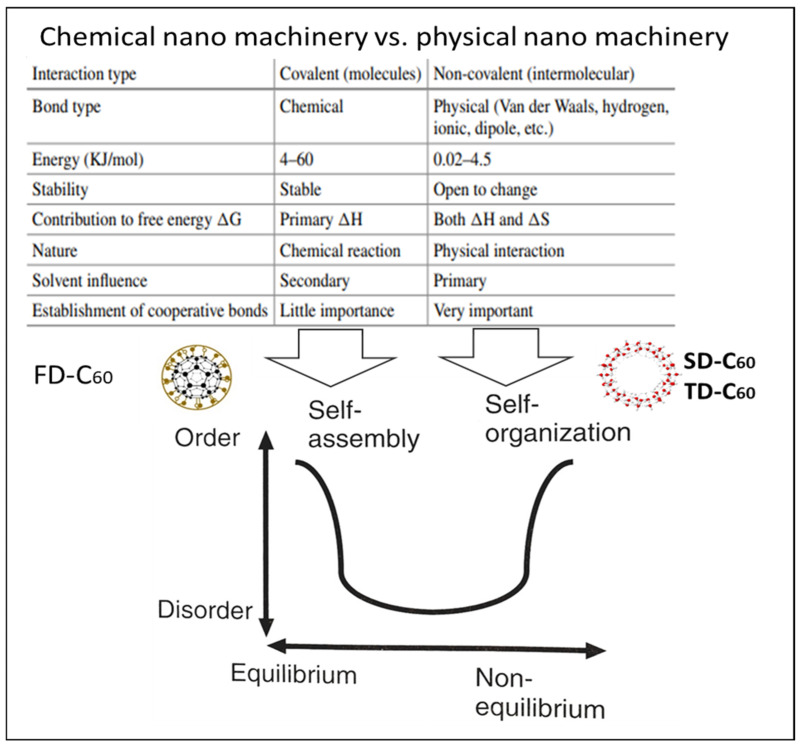
Presentation of similarities and differences of chemical and physical phenomena at the nano-level (Chemical nano-machinery vs. physical nano-machinery). Solvent influence is very important; secondary means it is not an active entity in the process, and primary means it is an active entity (Adapted from [23,24]). A similar situation exists in biology: protein synthesis, for example, tubulin, is a self-assembly process (biochemical), while the polymerization of tubulin into microtubules and microtubules into centrioles is a self-organization process (biophysical).

## Data Availability

The data presented in this study are available upon request from the corresponding author.

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
