# Peer review of "Molecular “Yin-Yang” Machinery of Synthesis of the Second and Third Fullerene C60 Derivatives"

_micromachines, 2025, doi:10.3390/mi16070770_

Round 1

Reviewer 1 Report

Comments and Suggestions for Authors

I would like to invite the respected Author to rewrite their manuscript in a coherent and clear scientific format. Respected Authors may find my comments as follows

1- Lack of central question. Can Authors elaborate on this in the introduction?

2- The abstract needs to be one section without sections. Respected authors should provide a coherent abstract.

3- Incomplete and out of standard sentences:  For instance, lines 23 and 24 of the abstract: "The paper explains the synthesis of SD-C60 and TD-C60 via hydrogen bonds ...." Do Authors try to explain the steps in their synthesis procedures or do they try to explain the mechanism of synthesis? In any case, such a sentence does not deliver any messages to the reader. The manuscript is full of these sentences. 

4- Non-standard figure captions, for instance, line 232-233, a quote from Linus Pauling appears in the figure caption.

5- Unexplained figure's part. Figure 3 has three parts, all need to be explained in the figure's caption 

6- Line 356-357: Respected Authors need to remove all such redundant sentences. Their work is about the synthesis of new fullerene-based nano-compound, it is not about the conceptual framework of quantum mechanics.  

Author Response

Please see the Attachment.

Regards.

Authors

Reviewer 2 Report

Comments and Suggestions for Authors

The research manuscript entitled Molecular “Yin-Yang” machinery of synthesis of the second and third fullerene C60 derivatives”. This work is systematically reported; however, it needs some clarification before acceptance. Comments are,

  1. In the abstract, recommended to rewrite with current challenges, research gaps, current research findings and conclusion.
  2. In abstract, the information about the magnetic field influences towards formation of C60 derivatives and their challenges to minimize the toxicity details are missing.
  3. In the introduction part, recommended presenting significant key points to enhance the manuscript quality.
  4. In fig.4b, 5a, TEM images particles scale range not clear
  5. In FT-IR results generally plotted wavenumber vs. Transmittance or absorbance, double check all the FT-IR spectra.

Author Response

Regards.

Authors

Round 2

Reviewer 2 Report

Comments and Suggestions for Authors

Authors improved the manuscript quality and clarified with adequate answers raised by the reviewer, recommended for publication